# Exploring Cognitive Deficits and Neuromodulation in Schizophrenia: A Narrative Review

**DOI:** 10.3390/medicina60122060

**Published:** 2024-12-14

**Authors:** Chien-Chen Hung, Ko-Huan Lin, Hsin-An Chang

**Affiliations:** 1Department of Psychiatry, Tzu Chi General Hospital, Hualien 970, Taiwan; s0987769818@gmail.com; 2Non-Invasive Neuromodulation Consortium for Mental Disorders, Society of Psychophysiology, Taipei 114, Taiwan; 3Department of Psychiatry, Tri-Service General Hospital, National Defense Medical Center, Taipei 112, Taiwan

**Keywords:** cognitive impairment, neuromodulation, schizophrenia, repetitive transcranial magnetic stimulation (rTMS), transcranial direct current stimulation (tDCS), transcranial alternating current stimulation (tACS), deep brain stimulation (DBS), electroconvulsive therapy

## Abstract

Cognitive deficits are emerging as critical targets for managing schizophrenia and enhancing clinical and functional outcomes. These deficits are pervasive among individuals with schizophrenia, affecting various cognitive domains. Traditional pharmacotherapy and cognitive behavioral therapy (CBT) have limitations in effectively addressing cognitive impairments in this population. Neuromodulation techniques show promise in improving certain cognitive domains among patients with schizophrenia spectrum disorders. Understanding the mechanisms of neural circuits that underlie cognitive enhancement is essential for elucidating the pathophysiological processes of the disorder, and these insights could significantly optimize strategies for managing schizophrenia. Meanwhile, although there is an increasing body of evidence demonstrating the therapeutic effects of neuromodulation in this area, further research is still needed, particularly regarding topics such as different treatment protocols and the long-term effects of treatment.

## 1. Introduction

Schizophrenia is a severe mental disorder affecting approximately 1% of the global population [1]. The symptoms of schizophrenia are diverse and debilitating. They are generally categorized as positive symptoms, such as hallucinations and delusions, and negative symptoms, like emotional flatness and social withdrawal. The repercussions of schizophrenia reach far beyond the individual, profoundly influencing families, communities, and healthcare systems alike. Schizophrenia imposes a significant economic burden [2], characterized by elevated healthcare expenses stemming from recurrent hospitalizations, ongoing medication requirements, and the demand for specialized mental health services.

Though not listed in formal diagnostic criteria (DSM-V), pervasive cognitive impairments are a core feature of schizophrenia. Cognitive deficits associated with schizophrenia may manifest before the initial acute episode and persist for decades [3]. While some studies suggest that cognitive impairment remains static throughout the disease trajectory [4], others indicate a gradual deterioration in the later stages [5]. Even in the remission state of positive symptoms, schizophrenic patients still struggle with persistent cognitive deficits and associated functional impairments [6]. The chronic progression of cognitive deficits in schizophrenia leads to increasing difficulty in personal care, social interactions, and employment, further isolating individuals with the disorder. Compared to other psychotic symptoms, cognitive impairment is more closely linked to prognosis and functional outcomes [7,8] and can result in prolonged institutionalization, heightening demand for mental health resources and contributing to a greater disease burden and increased economic costs [9].

Cognitive deficits in schizophrenia encompass a range of symptoms that may arise from specific brain region damage or more widespread functional disruptions within brain circuits. Traditional treatments for schizophrenia, primarily pharmacotherapy and cognitive behavioral therapy, have shown limited efficacy in addressing cognitive impairments. Consequently, neuromodulation targeting specific brain regions or circuits has been explored as a potential treatment for these symptoms. In this narrative review, we compile the relevant literature and systematically organize it to enable readers to grasp the key knowledge regarding cognitive impairment in schizophrenia effectively. We will begin by discussing the different domains of cognitive deficits in schizophrenia, linking various symptoms to specific brain regions and circuits. Subsequently, we will present existing evidence on neuromodulation as a treatment approach. Although some preliminary findings are promising, further research is necessary to substantiate these results.

## 2. Materials and Methods

We conducted a comprehensive literature search using PubMed to identify relevant studies published between 1995 and 2024. The search utilized keywords including schizophrenia, cognitive impairment, pharmacotherapy, specific brain circuits, specific brain regions, and specific neuromodulations. Our focus was directed towards several well-established brain regions and circuits that are known to be associated with cognitive deficits in schizophrenia. We included pertinent literature while systematically removing duplicate studies to ensure the integrity of our review. In organizing the literature, we first cataloged the various cognitive impairment symptoms and correlated these symptoms with specific brain regions. Subsequently, we compiled current evidence regarding neuromodulation approaches targeting cognitive impairments in schizophrenia, highlighting existing treatment evidence.

## 3. Domains of Cognitive Impairment in Schizophrenia

Cognitive impairments in schizophrenia are multifaceted and can be categorized into different domains, including attention, memory, executive function, social cognition, and perceptual–motor function. The importance of recognizing impaired cognitive domains is relevant to treatment.

### 3.1. Attention

Attention deficits are common in schizophrenia, with underlying factors such as difficulties in maintaining task focus and limitations in working memory. The study of attentional lapses and mind wandering in vigilance highlights the complexity of attention impairments in this disorder [10]. Brain imaging research has consistently shown that individuals with schizophrenia exhibit distinct patterns of brain activity compared to healthy controls during attention-related tasks. These differences are particularly evident in areas associated with attention, such as the dorsolateral prefrontal cortex (DLPFC), insula, anterior cingulate gyrus (ACG), amygdala, hippocampus, ventral striatum, thalamus, and cerebellum [11]. Our team found that lagged phase synchronization of high-frequency resting-state electroencephalography in the right hemispheric cuneus, superior temporal gyrus, and transverse temporal gyrus is associated with poor focused attention in stabilized schizophrenics [12]. This finding suggests a compensatory increase in functional connectivity among these structures.

### 3.2. Memory

Memory dysfunction in schizophrenia may have shared origins for deficits in working and episodic memory, as evidenced by prefrontal activation abnormalities [13]. Individuals with schizophrenia exhibit working memory deficits characterized by decreased performance accuracy and prolonged response times. Neuroimaging studies have shown dorsal frontal–parietal network dysfunction may be associated with working memory impairment. An inverted U-shaped activation of DLPFC has been linked to defected working memory in schizophrenics [14]. Episodic memory is often noted as exhibiting the most significant impact among the cognitive deficits observed in schizophrenia [15]. The characteristic episodic memory impairment in schizophrenia is marked by reduced accuracy, prolonged response times, deficient conscious recollection, failure of strategic encoding, and difficulty processing contextual information [16,17]. Deficits in episodic memory are likely best interpreted within the framework of a wider range of impairments in higher cognitive functions associated with schizophrenia (Sz). These deficits are often linked to dysfunction in the dorsolateral prefrontal cortex (DLPFC) and tend to become more apparent when there are significant demands for organization and cognitive control [13]. 

### 3.3. Executive Function

Executive function (EF) refers to mental abilities that help establish and achieve goals and encompasses a wide set of cognitive processes, including attention, working memory, decision-making, and mental flexibility. Deficits of executive function are prevalent in schizophrenia across stages, affecting tasks like conceptualization and planning [18,19].

Among various brain regions involved in executive function, including the limbic system and frontal cortex, the prefrontal cortex plays the most important role. The DLPFC is involved in working memory, reasoning, and thematic understanding, and the ventromedial prefrontal cortex (VMPFC) is involved in motivation and reward [20]. While the DLPFC is regarded as the most prevalent with executive dysfunction in schizophrenia, based on neuroimaging and neurophysiological studies [21], functional dysconnectivity between the prefrontal cortex and striatum, which is primarily involved in positive symptoms in schizophrenia, is highly associated with working memory impairment in schizophrenia [22]. Neuroimaging studies confirm dysfunction in the prefrontal cortex, which correlates with impairments in various neural networks. 

### 3.4. Social Cognition

Social cognition involves a multifaceted array of mental capacities that underpin the perception, processing, interpretation, and response to social stimuli. Collectively, these abilities facilitate the acquisition of appropriate social skills and adaptability. Impaired social cognition is a key feature of schizophrenia, affecting the ability to perceive, interpret, and respond to social interactions [23], significantly impacting interpersonal relationships, employment, and independent living. Social cognition is contributed by several different cognitive processes, including emotional recognition, intention recognition, and understanding of others’ perspectives and mental states. Hence, the brain regions involved in social cognition are extensive, including the amygdala, posterior superior temporal sulcus, temporoparietal junction, prefrontal medial region, and mirror neuron system involving inferior frontal gyrus, precentral gyrus, inferior parietal lobule, and temporal, occipital, and parietal visual areas [24,25]. 

### 3.5. Perceptual–Motor Function

Motor dysfunction is evident in schizophrenia [26]. Several mechanisms are postulated to be involved in motor dysfunction, including prediction errors, motor processing, and control [27]. The prediction–error hypothesis provides a framework to understand how the brain minimizes prediction errors to make adaptive decisions [28]. In schizophrenia, this theory suggests that disconnection in network connectivity can lead to the spontaneous generation of uncompensated prediction errors, which then causes unpredictable behavior [29]. The deficit of motor processing and control in schizophrenic patients is another significant aspect of the disorder [26]. Studies have consistently shown that patients with schizophrenia exhibit impairments in motor control, which can be attributed to abnormalities in various brain regions and neural systems, including the primary motor cortex, supplementary motor area, anterior cingulate cortex, prefrontal cortex, basal ganglia, and cerebellum [30,31].

## 4. Brain Circuits Involved in Cognitive Impairment in Schizophrenia

Cognitive impairment in schizophrenia is widespread, affecting various domains of cognition. Although different domains are primarily associated with specific brain areas, multiple brain circuits involving diverse brain structures play a crucial role in cognitive functions. In schizophrenia, cognitive impairment is closely linked to abnormalities in several key brain circuits. Dysconnectivity within and between specific brain regions and networks may contribute to cognitive deficits observed in schizophrenia. A concise visual representation of the various brain structures and networks involved in cognitive functioning in this disorder is illustrated in Figure 1.

### 4.1. Prefrontal Cortex (PFC)

Key regions of the PFC include DLPFC, VMPFC, medial prefrontal cortex (mPFC), and ACC. The PFC is essential for higher-order cognitive functions such as decision-making, problem-solving, and working memory [32]. In individuals with schizophrenia, there is often reduced gray matter volume in the PFC, which can affect these cognitive functions [33]. Working memory has also been considered a fundamental element of higher cognitive abilities [34] and functional magnetic resonance imaging study has shown that the dorsolateral prefrontal cortex (DLPFC) is involved in the defected working memory [35]. The VMPFC and mPFC are critical for self-referential thought and social cognition; abnormalities in these regions can result in deficits in self-awareness and perspective-taking, thereby exacerbating social isolation [36]. The anterior cingulate cortex (ACC) plays a crucial role in cognitive processes such as error detection, conflict monitoring, and attention regulation. Dysfunction within the ACC can lead to significant impairments in these cognitive functions, resulting in difficulties with decision-making and adaptive behavior [37]. This disruption in connectivity can significantly impact cognitive processes, including memory formation and retrieval, attentional control, and executive functioning. Specifically, alterations in PFC-hippocampal connectivity may contribute to deficits in working memory and learning, while changes in PFC-thalamic connections can affect the regulation of sensory information and attention. These connectivity disturbances are indicative of the broader network dysfunctions that underlie the cognitive impairments frequently observed in individuals with schizophrenia [38]. Dysregulation of key neurotransmitter systems, particularly dopamine and glutamate, further impairs PFC function, leading to deficits in executive functioning and cognitive control [39]. 

### 4.2. Hippocampal Circuitry

The hippocampus is critical for the formation and retrieval of memories, particularly episodic and spatial memory [40]. Reduced volume in the hippocampus is frequently observed in schizophrenia [41]. These structural abnormalities are linked to disrupted connectivity between the hippocampus and the PFC, contributing to difficulties in memory processes and spatial navigation [42]. This disruption can impair the ability to form new memories and recall past events accurately [13].

### 4.3. Thalamo-Cortical Circuitry

The thalamus functions as a relay station, processing and transmitting sensory information to the cortex and playing a role in consciousness regulation [43]. Schizophrenia is associated with abnormal thalamic volume and disrupted thalamo-cortical connectivity [44]. These alterations affect sensory gating mechanisms, which are crucial for filtering out irrelevant stimuli and focusing on important sensory inputs [45]. As a result, individuals with schizophrenia may experience sensory processing deficits and cognitive fragmentation, where thoughts become disjointed and incoherent [46].

### 4.4. Default Mode Network (DMN)

The default mode network (DMN), which includes regions such as the medial prefrontal cortex, posterior cingulate cortex, precuneus, and lateral parietal cortex, is most active during rest and is essential for self-referential thinking, mind-wandering, and introspection [47]. In schizophrenia, DMN connectivity is often disrupted, resulting in significant cognitive and social impairments [48]. This dysfunction affects tasks that require introspection, such as understanding one’s own thoughts and feelings, and theory of mind, which involves attributing mental states to oneself and others [49]. Consequently, individuals with schizophrenia struggle with social cognition, making it difficult to interpret and respond appropriately to social cues [23]. These impairments can manifest as inappropriate or intrusive thoughts during social interactions and contribute to difficulties in forming and maintaining relationships. Additionally, the inability to engage in effective self-reflection and social understanding exacerbates social withdrawal and isolation [50], further impacting the quality of life for those with schizophrenia.

### 4.5. Salience Network (SN)

The salience network (SN), which includes the anterior frontoinsular cortex, supplementary motor area, and anterior cingulate cortex, plays a crucial role in detecting and prioritizing important stimuli, thus aiding in focusing attention on relevant environmental cues [51]. In schizophrenia, dysregulated SN activity disrupts this filtering process, leading to an overwhelming influx of stimuli and causing attentional deficits and cognitive disorganization [52]. Neuroimaging studies highlight the SN’s consistent connectivity patterns essential for dynamic brain state transitions [53]. While dysconnectivity within SN structures, such as the supplementary motor area, has been reported to be related to global cognition [54], the modulation of the central executive network and DMN via anterior insula has been attributed to defected attention in schizophrenia [55]. Brain networks, including SN and DMN, are crucial in cognitive deficit in schizophrenia.

### 4.6. Central Executive Network (CEN)

The central executive network (CEN) is vital for cognitive functions, especially in the context of schizophrenia. This network, primarily located in the dorsolateral prefrontal cortex (DLPFC) and posterior parietal cortex, is essential for higher-order cognitive processes, including working memory, attention management, and decision-making. In individuals with schizophrenia, notable disruptions within the CEN have been identified, correlating with the cognitive impairments frequently observed in this disorder.

Recent investigations employing resting-state functional magnetic resonance imaging have uncovered abnormal connectivity patterns within the CEN and its interactions with other significant brain networks, particularly the default mode network (DMN) and the salience network (SN). Specifically, hypo-connectivity has been noted between the SN and both the DMN and CEN, suggesting a dysfunctional interplay that may contribute to cognitive deficits observed in schizophrenia [56].

### 4.7. Circuit Disruptions in Schizophrenia Animal Models

Animal models of schizophrenia have illuminated significant disruptions across several key brain circuits, including the prefrontal cortex (PFC), hippocampus, thalamo-cortical circuitry, default mode network (DMN), and salience network, all critical for cognitive, emotional, and sensory functions. The PFC, essential for executive function and working memory, exhibits reduced synaptic connectivity and dendritic spine density in models using NMDA receptor antagonists [57], mimicking hypofrontality seen in schizophrenia, while DISC1 mutant and COMT knockout models demonstrate impaired glutamatergic signaling and altered dopamine metabolism, leading to cognitive deficits [58,59]. The hippocampus, vital for memory and spatial navigation, shows disrupted long-term potentiation (LTP) and reduced connectivity with the PFC in neonatal ventral hippocampal lesion (NVHL) and NRG1 knockout models, paralleling schizophrenia-related memory deficits [60,61]. Thalamo-cortical circuits, which integrate sensory information and regulate attention, are impaired in NMDA antagonist and maternal immune activation (MIA) models, highlighting abnormal sensory gating and attentional processes [62,63]. DMN dysfunction, characterized by hyperconnectivity and reduced task engagement, is observed in prenatal stress models, linking developmental disruptions to attentional deficits [64]. Similarly, salience network dysfunction, critical for identifying and prioritizing stimuli, is evident in amphetamine-induced dopaminergic hyperactivity models, reflecting impaired anterior cingulate and insular activity [65]. Together, these findings underscore widespread neural disruptions in schizophrenia and provide a foundation for developing targeted therapeutic interventions.

## 5. Treatment for Cognitive Impairment

Conventional treatment for cognitive deficits in schizophrenia is a multifaceted approach that involves pharmacological interventions and behavioral therapies. Pharmacotherapy is the primary treatment for schizophrenia, with antipsychotic medications being the most widely used treatment [66]. Although antipsychotics had been proven to be effective for positive symptoms, they have limited beneficial efficacy on improving cognitive ability in schizophrenia [64,67]. Some studies even pointed out that antipsychotic agents might cause further cognitive impairment [68,69]. Atypical antipsychotics, compared to first-generation antipsychotics, have been shown to bring slight improvement in cognitive impairment in schizophrenia [70]. Cognitive enhancers, such as cholinesterase inhibitors and N-Methyl-D-aspartate receptor antagonists, have been investigated for their proven efficacy in cognitive deficit in neurodegenerative disorders [71]. However, a meta-analysis showed only a small significant effect size of cognitive enhancers on cognitive improvement [72]. On the other hand, cognitive behavioral therapy (CBT) has emerged as a promising adjunctive treatment for cognitive deficits in schizophrenia [73]. CBT focuses on improving cognitive skills through practice and learning strategies. Additionally, CBT can help individuals with schizophrenia develop better coping mechanisms and problem-solving strategies to manage their symptoms [74]. CBT has been shown to be effective on social impairment in schizophrenia [75] and used as an adjuvant treatment for symptom-related cognitive deficit in schizophrenia [74]. However, like psychotherapies of other modalities, CBT is subjected to labor-intensity, limiting its use in this aspect.

## 6. Neuromodulation

Neuromodulation techniques show promise for improving cognitive impairment in schizophrenia treatment. These techniques can regulate cortical excitability and neuroplasticity and promote brain connectivity [76]. On one hand, since different cognitive processes are controlled by unique brain areas, it is possible to enhance specific domains of cognitive function through a precise neuromodulation technique. On the other hand, cognition is complex and most cognitive processes are regulated by the interplay of multiple brain structures [77]. A broad-range and generalized brain stimulation might be helpful in this notion. A summary of different neuromodulations based on current evidence is listed in Table 1.

### 6.1. Repetitive Transcranial Magnetic Stimulation (rTMS)

Repetitive transcranial magnetic stimulation (rTMS) employs rapidly changing magnetic fields to generate electrical currents in targeted brain regions. This non-invasive technique involves placing a coil near the scalp through which an electric current passes to create a magnetic field that penetrates the skull and induces a secondary electric current in the brain tissue beneath the coil [78]. It can either excite or inhibit neural activity depending on the frequency and intensity of the magnetic pulses [78]. Compared with traditional suprathreshold brain stimulation, like electroconvulsive therapy (ECT), we can target rTMS on specific brain regions involved in different psychopathologies. High frequency rTMS brings about activation of the targeted brain area, and, by contrast, low frequency rTMS induces the suppression of the targeted brain area [79,80]. The treatment effect of rTMS is supposed to be mediated by modulation of the local brain areas and the functional connectivity between multiple involved brain structures [81].

This therapy has been applied to different psychiatric disorders [82]. In the context of cognitive deficit in schizophrenia, it is primarily applied to the DLPFC, a brain region crucial for executive functions, working memory, and other cognitive processes that are often impaired in individuals with this disorder [83]. High-frequency rTMS to the DLPFC also activates brain networks contributing to cognitive enhancement effect [84]. High-frequency rTMS applied to the DLPFC has shown potential in alleviating cognitive symptoms such as working memory deficits and executive dysfunction, which are common in schizophrenia and significantly impact patients’ daily functioning and quality of life [85]. A meta-analysis including nine clinical trials with sham controls showed high-frequency rTMS (left DLPFC and the total pulses < 30,000) has a significant acute effect on working memory and a significant long-term (2 weeks to 3 months) effect on working memory and language function in schizophrenic patients [86]. Intermittent theta-burst stimulation (iTBS), a variant protocol of rTMS which simulates the long-term potentiation in synaptic plasticity, showed a significant effect on social cognition when applied in the left DLPFC of patients with schizophrenia [87]. While traditional TMS is limited in effective electromagnetic field depth, approximately 2–3 cm beneath the sculp, deep TMS, utilizing combined and specialized coils, provides deep magnetic penetration and reaches deeper brain structures [88]. Hence, TMS is theoretically a powerful tool in enhancing subcortical regions and multiple brain networks. One pilot trial without controls including 10 schizophrenic patients found high-frequency deep TMS to prefrontal cortex improves executive function and sustained attention [89]. However, a subsequent double-blinded randomized controlled trial conducted by the same team found no significant effect of 20 Hz deep TMS on cognitive improvement in schizophrenia [90]. Further study is needed to verify the efficacy of deep TMS with different coils and treatment protocols. In general, rTMS has shown to have good safety and tolerability in multiple studies targeting patients with schizophrenia. The therapeutic potential of rTMS in cognitive deficit in schizophrenia is increasingly recognized, offering a promising complement to traditional pharmacological treatments [91]. Further research is crucial to fully comprehend the long-term benefits and optimize treatment protocols for individuals with schizophrenia.

### 6.2. Transcranial Electrical Stimulation (tES)

Transcranial electrical stimulation encompasses two distinct modalities: transcranial direct current stimulation (tDCS) and transcranial alternating current stimulation (tACS). The tDCS delivers a constant, low-intensity direct electric current through electrodes placed on the scalp, which modulates neuronal excitability [92]. This modulation depends on the polarity of the stimulation: anodal stimulation typically enhances cortical activity by depolarizing neurons and making them more likely to fire, while cathodal stimulation generally suppresses cortical activity by hyperpolarizing neurons and making them less likely to fire [92]. In the context of schizophrenia, tDCS is primarily applied to the DLPFC or other cortical areas implicated in cognitive processes [93]. Research has shown that tDCS can potentially improve these cognitive functions [94]. For instance, anodal stimulation of the DLPFC has been associated with enhancements in working memory and attention [95]. Some studies have also noted improvements in social cognition, which is critical for daily functioning and quality of life [96]. Our previous randomized sham-controlled trial found that schizophrenic patients who underwent tDCS experienced a rapid improvement in planning abilities and cognitive insight [97]. While initial results are promising, more research is needed to optimize tDCS protocols, determine the most effective stimulation parameters, and understand the long-term effects and safety of repeated tDCS sessions [98]. Additionally, further studies are necessary to identify which patients are most likely to benefit from this intervention and to elucidate the underlying mechanisms through which tDCS exerts its effects on the brain [99].

The tACS delivers low-amplitude biphasic electric currents to the scalp, which can modulate the activity of cortical neurons. This modulation is believed to occur through the entrainment of brain oscillations, potentially inducing long-term synaptic plasticity and enhancing cognitive functions and behavioral outcomes [100]. In our previous study, theta-frequency tACS improved working memory capacity of schizophrenia [101]. In addition, add-on theta-frequency tACS with pharmacotherapy improved working memory and processing speed in schizophrenia [102].

### 6.3. Deep Brain Stimulation (DBS)

Deep brain stimulation (DBS) represents a frontier in psychiatric treatment, involving the surgical insertion of electrodes into targeted brain regions critical to the pathophysiology of schizophrenia, such as the thalamus and hippocampus [103]. These electrodes deliver tailored electrical impulses, which can be adjusted for frequency and intensity, to modulate neural circuits that are dysfunctional in schizophrenia [104]. Compared to non-invasive brain stimulations such as rTMS and tDCS, DBS offers a direct stimulation over much deeper brain structures. Targeting brain regions and networks involved in cognitive function, this modulation has the potential to significantly enhance cognitive capabilities, including memory, attention, and executive function [105]. DBS has been tested for effectiveness on cognitive enhancement and the literature showed that DBS to fornix slows cognitive decline in patients with Alzheimer’s disease [106]. DBS’s capacity to reach deeper brain structures directly and adjust their activity makes it a particularly promising option for patients whose severe symptoms are resistant to pharmacological treatment [107]. However, there is currently a lack of evidence regarding the impact of DBS on cognitive deficits in schizophrenia. Further studies are needed to address this issue.

### 6.4. Electroconvulsive Therapy (ECT)

Electroconvulsive therapy (ECT) involves inducing controlled seizures through electrical currents administered to the brain [108]. While the precise mechanism of action is not fully understood, it likely involves a combination of neurochemical changes, neuroendocrine alterations, and enhanced neuroplasticity, which collectively contribute to its therapeutic effects [109]. In the context of schizophrenia, ECT is typically reserved for severe affective symptoms or cases that are resistant to other treatments [110]. Rajagopalan et al. reported cognitive improvement in patients with schizophrenia undergoing ECT. However, it remains challenging to ascertain whether the observed cognitive enhancement is directly attributable to the ECT treatment itself or if it is mediated by the improvement in negative or mood symptoms [111]. It is promising for ECT to ameliorate cognitive impairment linked to refractory psychotic symptoms.

### 6.5. Pharmacological Neuromodulation

Traditional schizophrenia treatments primarily target dopamine receptors to manage positive symptoms like hallucinations and delusions by blocking dopamine D2 receptors to reduce dopamine overactivity [112]. New treatments are investigating the roles of glutamate and acetylcholine in schizophrenia-related cognitive impairments [113]. Glutamate, the brain’s primary excitatory neurotransmitter, is crucial for cognitive processes, and dysregulation of glutamatergic neurotransmission, particularly NMDA receptor dysfunction, contributes to cognitive deficits in schizophrenia [114]. In response, novel pharmacological agents are being developed to target the glutamatergic system, aiming to correct hypoactivity in NMDA receptors [115]. These agents, including glycine site agonists and NMDA receptor co-agonists like D-serine, act as positive modulators of NMDA receptor function [116,117]. These molecules aim to improve cognitive deficits in schizophrenia by boosting NMDA receptor activity, correcting synaptic dysfunction, promoting synaptic plasticity, and restoring normal neural network function [118,119].

Moreover, the cholinergic system, including nicotinic and muscarinic acetylcholine receptors, is implicated in cognitive processes apart from its impact on positive symptoms [120,121]. Modulating these receptors offers a promising strategy, as nicotinic receptors enhance neurotransmission and synaptic plasticity, and muscarinic receptors influence cognitive pathways [122,123]. Effective treatments must boost cholinergic activity without causing anticholinergic effects, aiming to improve cognitive outcomes in individuals with schizophrenia. A summary of antipsychotics, including categories, mechanisms, advantages, and limitations, is presented in Table 2, offering a clear overview of pharmacological treatment options.

## 7. Conclusions

Our review underscores the critical importance of addressing cognitive deficits in schizophrenia, which pervasively and profoundly impact patients’ functional outcomes and quality of life. Cognitive impairments, affecting domains such as attention, memory, executive function, social cognition, and perceptual–motor functions, are core features of schizophrenia that remain inadequately treated by traditional antipsychotic medications.

The PFC, hippocampal circuitry, thalamo-cortical connectivity, DMN, CEN, and SN are all implicated in the cognitive deficits observed in schizophrenia. Understanding the intricate relationships and dysfunctions within these networks is essential for developing targeted neuromodulation interventions. While rTMS and tES have shown promise in enhancing working memory, attention, and social cognition, the efficacy and optimal protocols for these treatments require further investigation. DBS, though still experimental, and ECT, traditionally used for severe affective symptoms, also offer potential benefits for cognitive enhancement in some cases.

Pharmacological neuromodulation, focusing on the glutamatergic and cholinergic systems, presents additional strategies for cognitive improvement. Targeting NMDA receptor dysfunction and enhancing cholinergic activity are promising areas of research that could complement neuromodulation techniques and traditional pharmacotherapy.

In conclusion, the integration of neuromodulation techniques with existing treatments represents a significant advancement in the management of cognitive deficits in schizophrenia. Future research should focus on optimizing these interventions, understanding their long-term effects, and identifying patient subgroups most likely to benefit. By addressing cognitive impairments effectively, we can improve the clinical and functional outcomes for individuals with schizophrenia, ultimately enhancing their quality of life and reducing the overall burden of the disorder.

## Figures and Tables

**Figure 1 medicina-60-02060-f001:**
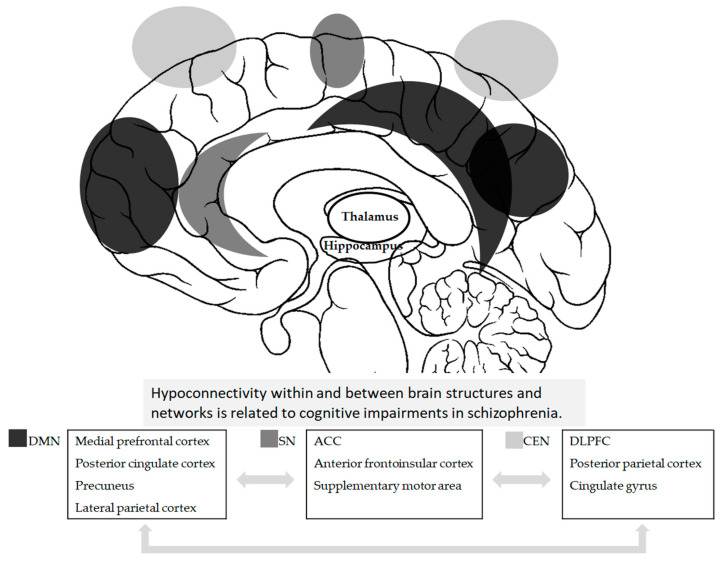
Diagrammatic representation of key brain regions and circuits involving cognitive impairment in schizophrenia. DMN = default mode network; SN = salience network; CEN = central executive network; ACC = anterior cingulate cortex; DLPFC = dorsolateral prefrontal cortex.

**Table 1 medicina-60-02060-t001:** Summary of different neuromodulations in treating cognitive impairment in schizophrenia.

	Target Region	Modality	Improved Cognitive Domains
rTMS	DLPFC	High-frequency rTMS iTBS	Working memory Language function Executive dysfunction Social cognition
tDCS	DLPFC	Anadol, 1–2 mA	Working memory Attention Social cognition
tACS	left frontoparietal areas	2 mA	Working memory Processing speed
ECT	Bitemporal Bifrontal Right unilateral (Electrode placement)		General (not specified)

rTMS = repetitive transcranial magnetic stimulation; tDCS = transcranial direct current stimulation; ECT = electroconvulsive therapy; DLPFC = dorsolateral prefrontal cortex; iTBS = intermittent theta-burst stimulation. Due to lacking evidence of deep transcranial magnetic stimulation and deep brain stimulation, the two treatments are not listed in this summary table.

**Table 2 medicina-60-02060-t002:** Summary of different antipsychotics in treating cognitive impairment in schizophrenia.

	Mechanism	Advantages	Limitations
FGAs	Block D2 receptors	-	High EPS, hyperprolactinemia risk, Minimal negative/cognitive efficacy
SGAs	Block D2, 5-HT2A, glutamate, histamine receptors	Lower EPS risk Improve negative symptoms, mood	Risk of metabolic syndrome Mixed cognitive efficacy
DRPAs	Partial D2/D3 agonists Balance dopamine activity	Lower EPS risk Improve negative symptoms, mood	Limited cognitive efficacy Akathisia
GMA	NMDA receptor co-agonist	Better for negative/cognitive symptoms than traditional antipsychotics	Inconsistent efficacy Often adjunctive
Emerging treatments	muscarinic, GABA, Sigma-1 receptors	Potential for cognitive symptoms improvement and modulating other deficits in schizophrenia	Early-stage trials with uncertain long-term safety and efficacy

FGAs = first generation antipsychotics (e.g., Haloperidol, Chlorpromazine, Fluphenazine); SGAs = second generation antipsychotics (e.g., Risperidone, Olanzapine, Quetiapine, Clozapine); DRPAs = Dopamine receptor partial agonists (e.g., Aripiprazole, Brexpiprazole, Cariprazine); GMA = Glutamate-modulating agents; Emerging treatments = Muscarinic receptor modulators (e.g., Xanomeline), GABAergic agents, Sigma-1 receptor agonists.

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
