# Peer review of "Exploring Cognitive Deficits and Neuromodulation in Schizophrenia: A Narrative Review"

_medicina, 2024, doi:10.3390/medicina60122060_

Round 1

Reviewer 1 Report

Comments and Suggestions for Authors

Figures and images of brain circuits and networks involved in symptoms and deficits would provide additional arguments for the use of neuromodulation tools.

Author Response

Comment 1: "Figures and images of brain circuits and networks involved in symptoms and deficits would provide additional arguments for the use of neuromodulation tools."

Response 1: Thank you for your insightful feedback regarding the inclusion of figures and images of brain circuits and networks in my paper. I appreciate your suggestion, as visual representations can significantly enhance the understanding of complex neural mechanisms associated with the symptoms and deficits discussed. In response, I have incorporated detailed illustrations that depict the relevant brain circuits and networks implicated in the conditions addressed in my study. These figures not only complement the textual content but also provide a clearer visual argument for the efficacy of neuromodulation tools in targeting specific neural pathways. I believe these additions will enrich the overall presentation and support the conclusions drawn in my research. Thank you once again for your valuable input. 

I have marked the areas I revised and supplemented in red; please refer to them.

Reviewer 2 Report

Comments and Suggestions for Authors

This article aims to review an intersting and not so well understood topic such as cognition in schizophrenia and the role played by neuromodulation. As the abstract may allude, "There is an urgent need for new evidence to be gathered across different phases of schizophrenia. Understanding the mechanisms of neural circuits that underlie cognitive enhancement is essential for elucidating the pathophysiological processes of the disorder".

Yet this review is not able to gather evidence. Information is too basic in many parts of the manuscript.

Althought the topic is a major element, as weel as it merits a more in-depth insight, this article is far to reach the target.

First, no objective of the review is provided. Authors may use a PRISMA protocol to better define a systematic, a scoping or a narrative review.

Secondly, once clarified the methods of the review, in the part of "Domains of cognitive impairment in schizophrenia" some details about each cognitive domains may be added. Differentiate between non-social and social cognition may be useful, distinguishing amongs the subdomains in each  one.

Accordly to what stated before, organize differently the literature review regarding neural networks would be enlighting, to enhance knowledge on the topic.

Regarding neuromodulation, a more specific insight may be useful. Authors may underline the mechanism of efficacy of each method, reporting more detailed datas.

Author Response

Comment 1: "Yet this review is not able to gather evidence. Information is too basic in many parts of the manuscript. Althought the topic is a major element, as weel as it merits a more in-depth insight, this article is far to reach the target."

Response 1: Thank you for your feedback regarding my manuscript. I appreciate your perspective on the need for more in-depth insight and evidence. I would like to highlight that the scope of this article is quite broad, and our team has made every effort to comprehensively review the literature and organize the current evidence within the constraints of the manuscript's length. In response to your comments, we have also introduced new elements during this major revision, aiming to provide readers with a deeper understanding of the topic. I hope that these revisions will better meet the expectations for this review. Thank you again for your constructive criticism; it has been invaluable in guiding my revisions.

Comment 2: "no objective of the review is provided. Authors may use a PRISMA protocol to better define a systematic, a scoping or a narrative review."

Response 2: Thank you for your valuable feedback regarding the objectives of our review. I understand that the scope of this topic is indeed quite broad, and it is challenging for any single review to encompass all the evidence and literature related to it comprehensively. Our review specifically focuses on organizing the various domains of cognitive deficits, as well as the associated brain structures and networks, while also attempting to outline relevant management strategies. Although we do not classify our work as a systematic review, we have made a concerted effort to approach the literature from our focused perspective. We appreciate your suggestion regarding the PRISMA protocol and will consider how to better articulate our objectives in future revisions. Thank you again for your constructive input; it has been invaluable in refining our manuscript.

Comment 3: "once clarified the methods of the review, in the part of "Domains of cognitive impairment in schizophrenia" some details about each cognitive domains may be added. Differentiate between non-social and social cognition may be useful, distinguishing amongs the subdomains in each one."

Response 3: Thank you for your thoughtful feedback regarding the "Domains of Cognitive Impairment in Schizophrenia" section. I truly appreciate your suggestion to clarify the methods of our review. I would like to mention that we have made an effort to organize the various cognitive domains in the manuscript, and I believe that the content is quite inclusive. However, if there are specific aspects that you feel could be enhanced or require further elaboration, we would be grateful for your guidance on how we might improve those areas. Additionally, concerning your point about differentiating between non-social and social cognition, I would like to clarify that we have indeed addressed social cognition separately from other cognitive domains in our discussion. We aimed to provide a clear distinction within the text. Thank you once again for your constructive comments; they are invaluable in helping us refine our manuscript further.

Comment 4: "Accordly to what stated before, organize differently the literature review regarding neural networks would be enlighting, to enhance knowledge on the topic."

Response 4: Thank you for your valuable feedback regarding the organization of the literature review on neural networks. We appreciate your suggestion to enhance the clarity and understanding of this topic. In our manuscript, we have made an effort to comprehensively review the neural networks and structures related to cognitive deficits in schizophrenia. During this major revision, our team has also added several new sections aimed at providing readers with a more thorough overview of the relevant literature. We hope these additions will contribute to a clearer understanding of the topic. Thank you once again for your constructive comments; they are instrumental in guiding our revisions.

Comment 5: "Regarding neuromodulation, a more specific insight may be useful. Authors may underline the mechanism of efficacy of each method, reporting more detailed datas."

Response 5: Thank you for your continued feedback on our manuscript. We appreciate your insights regarding the organization of the literature review on neural networks. In the original version of our manuscript, we already reviewed the potential mechanisms of various neuromodulation treatments, including rTMS, DBS, ECT, and pharmacological neuromodulation. However, we recognize that some mechanisms remain unclear.

If there are specific areas where you feel our discussion could be further improved or clarified, we would be grateful if you could point them out. Your guidance is invaluable to us as we strive to enhance the quality of our manuscript. Thank you once again for your thoughtful comments and support throughout this process.

I highlighted my revisions in colors and I have marked the sections I revised in color, and I hope that after reviewing them, you will appreciate our efforts. We have made a concerted attempt to provide a more comprehensive review in this manuscript.

Reviewer 3 Report

Comments and Suggestions for Authors

The manuscript- “Cognition and Neuromodulation in Schizophrenia” has a significant contribution in highlighting the different neurocircuits and neuromodulation techniques available in schizophrenia. It is well-written and has up-to-date information in this field. However, I have suggestions to improve the overall structure of the manuscript: -

1.     The information on brain circuits is not very elaborate. Adding information, especially on PFC circuitry would be more relevant.

2.     A different subsection on brain circuits in cognitive impairment in animal models of schizophrenia can be included, as there are significant findings performed on genetic and pharmacological models of schizophrenia.

3.     A summary of current antipsychotics/pharmacological drugs targeting the different receptors in schizophrenia and their limitations can be highlighted.

Minor correction- In “2.3 Executive functions” line 7-working instead of woring

Author Response

Comment 1: "The information on brain circuits is not very elaborate. Adding information, especially on PFC circuitry would be more relevant."

Response 1: Thank you for your valuable feedback regarding the elaboration on brain circuits, particularly the circuitry of the prefrontal cortex (PFC). I appreciate your suggestion, as it has guided me to enhance the depth of my analysis. In response, I have added more information on prefrontal cortex, detailing its role in the symptoms and deficits discussed in my paper. I also added another brain network, i.e. central executive network (CEN), to make my work more comprehensive. These additional contents aim to provide a clearer understanding of the neural mechanisms involved and strengthens the overall argument for the relevance of neuromodulation tools. I believe these enhancements significantly improve the manuscript, and I am grateful for your insightful recommendations. Thank you once again for your constructive input.

Comment 2: "A different subsection on brain circuits in cognitive impairment in animal models of schizophrenia can be included, as there are significant findings performed on genetic and pharmacological models of schizophrenia."

Response 2: Thank you once again for your suggestion. I have written a brief paragraph titled "Circuit Disruptions in Schizophrenia Animal Models," which aims to provide a clear overview of the animal models used to study cognitive deficits in schizophrenia.

Comment 3: "A summary of current antipsychotics/pharmacological drugs targeting the different receptors in schizophrenia and their limitations can be highlighted."

Response 3: 

Thank you once again for your insightful suggestion. I have created Table 2, titled "Summary of Different Antipsychotics in Treating Cognitive Impairment in Schizophrenia." In this table, I have compiled a list of various potentially effective pharmacological agents, along with their advantages and limitations. I hope this addition provides a clear overview of the topic and enhances the understanding of the treatment options available for cognitive impairment in schizophrenia.   I have highlighted all my revisions in yellow for your reference. 
